# Drought resistance of ten ground cover seedling species during roof greening

**Pengqian Zhang[1,2☯], Jiade Bai[1☯], Yanju Liu[3]\*, Yuping Meng[1], Zheng Yang[1], Tian Liu[1]**

**1** Beijing Biodiversity Conservation Research Center, Beijing, China, **2** Beijing Gardening and Greening Bureau, Beijing, China, **3** Beijing Center for Physical and Chemical Analysis, Beijing, China

☯ These authors contributed equally to this work.
\* liuyanju@hotmail.com

## Abstract

Roof greening is an important national policy for maintaining the hydrological balance in China; however, plant growth is limited by drought stress. This study aims to identify strong drought resistant plant species for roof greening from ten common species: *Paeonia lactiflora*, *Hemerocallis dumortieri*, *Meehania urticifolia*, *Iris lactea* var. *chinensis*, *Hylotelephium erythrostictum*, *Sedum lineare*, *Iris germanica*, *Cosmos bipinnata*, *Hosta plantaginea*, and *Dianthus barbatus*. By controlling the soil relative water content (RWC), we designed three treatments: moderate drought stress (40±2% < RWC < 45±2%), severe drought stress (RWC < 30±2%) and well-watered control (RWC > 75±2%). After the seedlings were provided different levels of water, their membrane permeability (MP), chlorophyll concentration (Chl), and superoxide dismutase (SOD), peroxidase (POD) and ascorbate peroxidase (APX) activity were measured. Finally, the membership function method was used to assess the drought resistance of these species. The results showed that *C. bipinnata* and *M. urticifolia* were not suitable for moderate or severe drought stress and did not survive. The other species presented variations in physiological and biochemical parameters. The MP of *He. dumortieri*, *I. lactea* and *Ho. plantaginea* showed minor changes between the well-watered control and drought stress. Most of the species showed reduced SOD activity under moderate drought stress but increased activity under severe stress. All of the plant species showed decreases in the protective enzymes POD and APX with increasing drought stress. The membership function method was applied to calculate the plant species' drought resistance, and the following order of priority of the roof-greening plant species was suggested: *He. dumortieri* > *I. germanica* > *I. lactea* > *D. barbatus* > *Hy. erythrostictum* > *S. lineare* > *Ho. plantaginea* > *P. lactiflora*.

**Data Availability Statement:** All relevant data are within the paper and its supporting information files.

## 1 Introduction

Roof greening, which is regarded as the "fifth surface greening", is one of the fundamental measures for sponge cities and represents an important national policy for improving the relationship between city development and nature protection to maintain the hydrological balance in China [1,2]. As an important supplement of urban landscaping, roof greening can help

**Funding:** This work was supported by the Young Core Plan of the Beijing Academy of Science and Technology (BJAST) (No. 201528), the Beijing Natural Science Foundation (No. 8142017) and the National Natural Science Foundation of China (No. 41475133).

**Competing interests:** The authors have declared that no competing interests exist.

mitigate the urban heat island effect [3], improve air quality [4] and enrich the biodiversity of cities [5]; hence, this landscaping style has expanded throughout all of China. Green roofs can be categorized roughly into two types: those that consist of diverse types plants (shrubs, trees, grasses, and flowers), namely, intensive green roofs (IGRs), and those that consist of simple herbaceous plant species, namely, extensive green roofs (EGRs) [6]. To grow on roofs, plants face many challenges. Taking Beijing as an example, plants that compose green roofs suffer from restricted rainfall in winter, spring, and autumn and evaporation always increases with high summer temperatures. Drought is considered as one of the most common environmental stresses that currently affects plant growth [7,8]. When plants experience drought stress, reactive oxygen species (ROS) are produced [9] including singlet oxygen ($^1O_2$), superoxide radical ($O_2^-$), hydroxyl free radical ($\bullet OH$), and hydrogen peroxide ($H_2O_2$) [10]. ROS can reduce crop productivity and plant viability because they can cause oxidative damage to proteins, DNA, and lipids [11]. Accordingly, drought stress can not only disrupt leaf membrane permeability (MP) [12] but also reduce the chlorophyll concentration (Chl) [13] and the activity of superoxide dismutase (SOD) [14], peroxidase (POD) [15] and ascorbate peroxidase (APX) [16]. These indicators are used for measuring the degree of plant drought stress, and they are usually analyzed as a whole due to their close associations, such as the ability of antioxidative enzymes SOD, POD and APX [17] to quench ROS and protect the cell from damage.

Biological membranes are crucial aspects of living systems that control the organization and distribution of different chemical components [18], and maintain sufficient water in plant tissue to protect the organism from dehydration and carboxylation and prevent enzymes from inactivating [19]. Liposomes are colloid vesicles composed of a lipid bilayer membrane and a watery internal compartment [20], and they serve as transport carriers for the efflux of secreted proteins. Low temperature [21], drought [22], salt [23] and heavy metal [24] stress break the stability of the plant cell membrane system and proteins, thereby increasing biofilm fluidity, altering the conformation of proteins, and then leading to physiological, biochemical and metabolic imbalance and abnormalities [25]. MP is determined by electrolyte leakage [26] and could be estimated by measuring electrolytes seeping from the plant cells under environmental stress. Generally, the greater the value of MP, the more cell damage there is. Under the same water condition, a lower MP value implies a stronger adaptation of the plant species to the environment.

Chlorophyll, a green pigment, is widely distributed in plant leaves and stems [27]. It helps convert absorbed solar radiation into stored chemical energy [28] and binds to proteins within chloroplasts and affects the light-harvesting capability and photosynthesis of plants[29, 30]. Upon drought stress, plant Chl is mainly affected by the physical destruction of chloroplasts and the inhibition of Chl a and Chl b functionality. Drought stress also causes the chloroplast matrix lamella to bend and swell [31], thereby impeding Chl synthesis and reducing its production [32]. In addition, reactive oxygen species (ROS, $^1O_2$, $O_2^-$ and $\bullet OH$) can directly or indirectly lead to lipid peroxidation and thus Chl damage [33].

The antioxidant system in plants consists mainly of nonenzymatic antioxidants and antioxidant enzymes. The most important antioxidant system in plants is composed of the antioxidant enzymes in chloroplasts and the cytoplasm [34]. SOD is an important enzyme that is ubiquitously expressed in aerobic organisms and catalyzes the dismutation of superoxide anions to hydrogen and molecular oxygen, which constitutes the first line of defense against ROS at the cellular level [35, 36]. Based on the prosthetic metal at the active site, SODs are classified into three groups, namely, CuZn-SODs, Mn-SODs, or Fe-SODs [37], of which Mn-SODs are closely related to mitochondria [38] and CuZn-SODs are mainly located in the cytoplasm and chloroplasts of plant cells [39]. McCord and Fridovich [40] described the principle chemical reaction under the elimination of ROS by SODs as $O_2^- + O_2^- + 2H^+ \rightarrow O_2 + H_2O_2$.

Existing in peroxisomes, glyoxysomes, vacuoles, the nucleus, and the extracellular matrix, SODs play a critical role in drought tolerance [41]. The SOD activity reflects the ability of plant species to adapt to environmental stress. Higher SOD activity values represent a stronger adaptation ability [42].

The antioxidant enzyme POD can scavenge and breakdown ROS [43] via the reaction $RH_2+H_2O_2 \rightarrow 2H_2O+R$, in which $H_2O_2$ is thoroughly converted into $H_2O$ [44, 45]. Chen *et al* [46] and Wu *et al* [47] found that increased POD activity helped cucumber and *Dendrobium moniliform* alleviate oxidative damage under drought stress. POD can further scavenge peroxides induced by SOD, and the synergistic action between these enzymes constitutes the protective enzyme system of the organism. Changes in the activity of these enzymes under stress may reflect the plant resistance ability in adverse environments.

APX is a member of the class I heme peroxidases and an important enzyme in plant antioxidant defense systems, and APX with several isoenzymes has a strong ability to scavenge ROS [48–51]. APX has been found in most eukaryotes, including higher plants[52], where it plays a key role in the metabolism of $H_2O_2$. Stronger APX activity would more quickly remove $H_2O_2$, thus preventing oxidative damage [53]. Different kinds of POD isoenzymes have obvious tissue and organ specialization. Similarly, APX is distributed in chloroplasts and the cytoplasm. POD and APX differ in their composition, structure, substrate specialization affinity, and stability during the purification process [54].

The membership function method is widely used to assess plant stress resistance. For example, it is has been used to assess the drought tolerance of *Malus* [55], maize [56] and potato [57] and the salt tolerance of *Sorghum bicolor* [58], *Lactuca sativa* [59], sugar beet [60], etc. The membership function weighted average method (*D* value) not only eliminates the one-sidedness associated with individual indexes but is also a relatively reliable evaluation method because the *D* value is the pure number within the closed interval of [0,1], which makes the difference in drought resistance of each test material comparable [61].

According to "Beijing local standards, roof greening specification (DB11/T 281–2005)" [62], more than 20 species can be used for ground cover and roof greening. Studies have focused on most of these species for their drought resistance, although these studies were limited to one species or one family. Because of the lack of studies comparing the drought resistance between these species, this study aims to screen plant species with strong resistance under drought stress to provide government policymakers with scientific plant species choices to improve plant survival rates and save maintenance costs during roof greening.

## 2 Study area overview

Milu Park is located 2 km far away from the South 5$^{th}$ Ring Road in Beijing and surrounded by Nan-Haizi Suburb Park. The drought stress experiment was conducted under a rain shed in the core-protection area for David's Deer in Milu Park (39.78˚N, 116.47˚E). During the test period, the daily average temperature was approximately 18.7˚C, the daily average humidity was approximately 55.2%, and the daily average illumination intensity (at 12:00) was approximately 2000 lx. The experiment was performed in the middle of April to the end of May 2015.

## 3 Materials and methods

This research has been held in the Beijing Milu Ecological Research Center (also known as Milu Park), located in Daxing district, Beijing, China. Milu Park is a place dedicated to ecological science research, as well as offered popular science education for the public for free. The authors, as staffs of Milu Park, in charge of conducting scientific research including biological science and environmental science. No additional permission is required for the authors to

carry out the experiments here. Also, the 10 plant species used for experiments were all market-purchased, common ground cover plants. These plants are not endangered rare plants, not be protected.

### 3.1 Seedlings

One-year-old seedlings of ten species, i.e., *Paeonia lactiflora*, *Hemerocallis dumortieri*, *Meehania urticifolia*, *Iris lactea* var. *chinensis*, *Hylotelephium erythrostictum*, *Sedum lineare*, *Iris germanica*, *Cosmos bipinnata*, *Hosta plantaginea*, and *Dianthus barbatus*, were provided by the Yu-Quanying flower market, a large and popular wholesale market in Fengtai District, Beijing that supplies most ornamental plants for Beijing City. The plant species present various propagation modes and other characteristics (Table 1).

### 3.2 Field soil collection and preparation

The field soil was collected from a wild wetland area in Milu Park, and it was then air-dried and ground to powder for the transplantation experiment. The soil was moderately saline (pH = 7.89) and presented available nitrogen, available phosphorus, and available potassium contents of 24.7 mg·kg$^{-1}$, 18.9 mg·kg$^{-1}$, and 322 mg·kg$^{-1}$, respectively [63].

### 3.3 Plant transplanting

The transplantation program was as follows: first, approximately 400 g powdered soil was placed in a plastic pot that was 20 cm tall and 13 cm in diameter and had 3 small holes at the bottom for drainage. Second, after removing the plastic wrap surrounding the roots, the seedlings were carefully planted at the pot's center. Third, another approximately 300 g of powdered soil was placed into the pot to cover the roots and then compressed tightly by hand. The seedlings were watered every 10 min, three times in total, to ensure that enough water was available to support plant growth. The transplantation was a success if new leaves and fresh stems were developed. The seedling survival rate reached 99% one week after replanting.

### 3.4 Drought stress treatment design

Three treatments were designed for each of the ten species: two drought stress level treatments, which included moderate drought stress (MDS or moderate; the water content in the soil

**Table 1. Plant species and their propagation modes together with other basic features.**

| Latin name | Propagation mode | Life cycle | Family | Species characteristics |
|---|---|---|---|---|
| *Paeonia lactiflora* | division of suckers | perennial | Ranunculaceae | Popular in gardening, and roots used as traditional Chinese medicine |
| *Hemerocallis dumortieri* | sowing of seeds | perennial | Liliaceae | Native to Northeast China, North Korea, Japan and Russia |
| *Meehania urticifolia* | sowing of seeds | annual or perennial | Lamiaceae | Adapted to dark and moist environments |
| *Iris lactea* var. *chinensis* | sowing of seeds | perennial | Iridaceae | Tolerant to saline-alkaline conditions and presents a well-developed root system |
| *Hylotelephium erythrostictum* | cuttings of seeds | perennial | Crassulaceae | Traditional Chinese medicine |
| *Sedum lineare* | sowing of seeds | perennial | Crassulaceae | Traditional Chinese medicine |
| *Iris germanica* | rhizome cuttings | perennial | Iridaceae | Native to Europe |
| *Cosmos bipinnata* | sowing seedling | annual or perennial | Asteraceae | Native to Mexico |
| *Hosta plantaginea* | division of suckers | perennial | Liliaceae | Traditional Chinese medicine |
| *Dianthus barbatus* | sowing of seeds | perennial | Caryophyllaceae | Native to Europe |

varied from 40±2% ~ 45±2%) and severe drought stress (SDS or severe; the water content in the soil was less than 30±2%), and one control group (CG or well-watered), which was under sufficient soil water conditions (the water content in the soil was over 75±2%) [64, 65]. For each treatment, three replicates were performed. Drought stress was dependent on natural evaporation. During the drought stress period, a WET-2™ sensor made by Delta-T Devices, Ltd., Cambridge, UK, was applied to measure the water content. Once the relative water content (RWC) of the soil met the requirements of the experiment, the plant seedlings were maintained under those conditions for approximately two days to ensure that changes in plant physiology and biochemistry had occurred. For the well-watered treatment, the seedlings were watered every four days.

## 3.5 Leaf sampling

All plants grew new leaves ten days after transplanting, which indicated the plants' roots had developed by the time leaves could react to the plant's physiological status. Referring to the sampling method in VDI-Guideline 3975 Part 11 [66], at least 15 g of healthy leaves was collected for each replicate. The leaf samples were placed into sealed plastic bags under a portable ice-box at 0~4˚C before being transferred to the lab for further physiological and chemical analysis.

## 3.6 Determining the MP, Chl, SOD activity, POD activity, and APX activity

The MP (%) of the leaves was calculated as $MP = \frac{L_t - L_{Cg}}{1 - L_{Cg}} \times 100$, where $L_t$ is the relative electrical conductivity of the plant material in the drought stress treatments and $L_{Cg}$ is the relative electrical conductivity of the material in the control group. The relative electrical conductivity $L = \frac{S_1 - S_0}{S_2 - S_0}$, where $S_1$ is the original conductivity of the deionized water with fractured fresh leaves, $S_2$ is the conductivity of the boiled deionized water with fractured leaves, and $S_0$ is the conductivity of deionized water [67]. The leaf MP was determined using a Thermo Scientific™ Orion 3-star inductivity- measuring device. Before the test, all sample leaves were flushed with deionized water 3 times and residual water on the leaf surface was removed by absorbent paper.

Chl was estimated according to the method described by Arnon [68] and Zhang *et al* [69] in detail. Three grams of fresh leaf material was crushed with a mortar and extracted with 10 mL of 80% acetone for 15 min. The extracted solution was then centrifuged at 2500 rpm (F = 34.9 g) for 3 min and measured at wavelengths of 643 nm, 645 nm, and 663 nm via a spectrophotometer (Metash™ UV-6100A). Calculations were performed via the formulas below.

$$\text{Chlorophyll a } (\text{Chla}, \text{mg} \cdot \text{L}^{-1}) = 12.7\text{A}_{643} - 2.69\text{A}_{645}$$

$$\text{Chlorophyll b } (\text{Chl b}, \text{mg} \cdot \text{L}^{-1}) = 22.9\text{A}_{645} - 4.68\text{A}_{663}$$

$$\text{The total chlorophyll of the solution } (\text{C}_\text{T}, \text{mg} \cdot \text{L}^{-1}) = \text{Chl a} + \text{Chl b}$$

$$\text{Chlorophyll concentration } (\text{Chl}, \text{mg} \cdot \text{g}^{-1}) = \text{C}_\text{T} * \text{V}/\text{W}/1000$$

where Chl a and Chl b refers to the concentration of chlorophyll a and chlorophyll b of the extracted solution; $\text{A}_{643}$, $\text{A}_{645}$ and $\text{A}_{663}$ refer to the absorbance of the measured solution at wavelengths of 643 nm, 645 nm and 663 nm, respectively; $\text{C}_\text{T}$ (mg/L) is the total chlorophyll of the solution; V represents the total volume of the extracted solution (mL); and W is the weight

**Table 2. Solution for the SOD, POD, and APX reaction systems.**

| Protective enzyme | Solution | mL | Final concentration |
|---|---|---|---|
| SOD | Phosphate buffer (0.05 mol·L$^{-1}$) | 1.5 | - |
| | Met solution (130 mmol·L$^{-1}$) | 0.3 | 13.0 mmol·L$^{-1}$ |
| | Nitro blue tetrazolium solution (750 μmol·L$^{-1}$) | 0.3 | 75.0 μmol·L$^{-1}$ |
| | EDTA-Na$_2$ solution (100 μmol·L$^{-1}$) | 0.3 | 10.0 μmol·L$^{-1}$ |
| | Riboflavin solution (20 μmol·L$^{-1}$) | 0.3 | 2.0 μmol·L$^{-1}$ |
| | Crude enzyme | 0.1 | Illumination check replaced with phosphate buffer |
| | Distilled water | 0.5 | - |
| | Total volume | 3.3 | - |
| POD | Phosphate buffer (0.05 mol·L$^{-1}$) | 2.9 | - |
| | H$_2$O$_2$ (2%) | 0.5 | - |
| | 2-Hydroxyanisole solution (2%) | 0.1 | - |
| | Crude enzyme | 0.1 | - |
| APX | Phosphate buffer (pH = 7.0, 0.05 mol·L$^{-1}$) | 1.8 | - |
| | Ascorbic acid solution | 0.1 | - |
| | H$_2$O$_2$ (0.3 mmol·L$^{-1}$) | 1.0 | - |
| | Crude enzyme | 0.1 | - |

of the extracted leaf (g). In the final result, Chl (mg·g$^{-1}$) refers to the chlorophyll content contained within each gram of leaf sample.

Crude enzyme extracts from the leaves were used to measure the SOD, POD, and APX activity. Approximately 0.5 g fresh leaves was added with a slight amount of CaCO$_3$, high-purity quartz sand and 5 mL of phosphate buffer (0.05 mol·L$^{-1}$) and then crushed into a powder in a mortar under freezing conditions. The mixture was subsequently transferred to a 10 mL centrifuge tube and then diluted with deionized water to 10 mL. The samples were then centrifuged at high speed (F = 13000 g) for 20 min at 0~4˚C [67].

The SOD and POD reaction systems were established as described by Zhang *et al* [67], and the APX reaction system was described by Tang *et al* [70] as shown below:

Two copies of the reaction system solution for each leaf sample were configured following Table 2-SOD mentioned above, one of which was put into a test tube and illuminated with 4000 lx for approximately 20~30 min at room temperature, the other one was put into the check tube wrapped in aluminum foil to avoid illumination. Once the color of the solution transition started, the reaction was immediately stopped. The final solution absorbance value was determined with a Metash™ UV-6100A spectrophotometer at a wavelength of 560 nm.

$$\text{SOD activity } (\text{U} \cdot \text{mg}^{-1}) = \frac{(A_0 - A_S) \times V_T}{A_0 \times 0.5 \; W \times V_1} \times dilution \; ratio$$

where $A_0$ is the absorbance of the check tube solution; $A$s is the absorbance of the test tube solution; $V_T$ (mL) is the total volume of the samples; $V_1$ (mL) is the volume of the reaction system; and $W$ (g) is the weight of the fresh leaves.

All the components were put into a test tube, and then the components of the POD reaction system were added (Table 2-POD). The solution's light absorption value was recorded for each tube (the wavelength was maintained at 470 nm). The solution was read every 1 min, and each solution was recorded 5 times in 5 min.

$$\text{POD activity } (\text{U} \cdot \text{g}^{-1} \cdot \text{min}^{-1}) = \frac{\Delta A_{470} \times V_T}{W \times V_s \times 0.01 \times t}$$

where $\Delta A_{470}$ is the change in absorbance during the reaction period, $W$ (g) is the weight of the sample, $t$ (min) is the reaction time, $Vs$ (mL) is the volume of the reaction system, and $V_T$ (mL) is the total volume of the sample.

With respect to the APX reaction system (Table 2-APX), the mixture was put into a test tube, after which the light absorption values were recorded at 290 nm every minute; this step was repeated 5 times. The formula to calculate the APX activity was as follows:

$$\text{APX activity } (U \cdot min^{-1} \cdot g^{-1} FW) = \frac{\Delta A_{290} \times V_1}{0.01 \times V_2 \times t \times W}$$

where $\Delta A_{290}$ is the change in absorbance during 5 min, $V_1$ (mL) is the volume of the crude enzyme, $V_2$ (mL) is the volume of the crude enzyme involved the reaction (0.1 mL in this test), $t$ (min) is the reaction time (5 min in this test) and $W$ (g) is the weight of the fresh leaves. FW is short for fresh weight.

### 3.7 Data analysis

SPSS 17.0 and Excel 2010 for Windows were used to calculate the mean, SD, etc. Multiple comparisons of the means by the least significant difference (Tukey's honestly significant difference [HSD]) test were performed on the 5 parameters (MP, Chl, SOD activity, POD activity, and APX activity) under the two drought stress treatments and the control group. ANOVA was used to determine significant differences between 10 species and between three treatments ($P<0.05$).

Following a drought resistance assessment method [71] for plant species based on the membership function value in fuzzy mathematics was used, the MP, Chl, SOD activity, POD activity, and APX activity results can be integrated into a single value for each species. The membership function value was calculated as follows:

$$\widehat{X}_{ij} = \frac{X_{ij} - X_{imin}}{X_{imax} - X_{imin}} \qquad (1)$$

$$\widehat{X}_{ij} = 1 - \frac{X_{ij} - X_{imin}}{X_{imax} - X_{imin}} \qquad (2)$$

where the lowercase "i" and "j" represent the plant species and the parameter type, respectively; "$\widehat{X}_{ij}$" is the mean value of the parameter "j" of the species "i"; "$X_{imax}$" and "$X_{imin}$" represent the maximum and minimum of the parameter "j" of the "i" species; and "$\widehat{X}_{ij}$" is the membership function value and represents the drought resistance of the seedlings. The average of the membership function value was then applied to estimate the adaptive capability of the plants under drought stress. After calculation according to formula (1) or (2), only positive "$\widehat{X}_{ij}$" value was chosen as the result. The formula for the average was as follows (where "$n$" represents the number of parameters, and "$\overline{X}_i$" represents the average "$\widehat{X}_{ij}$"):

$$\overline{X}_i = \sum \widehat{X}_{ij}/n$$

## 4 Results and discussion

### 4.1 Soil relative water content

The soil relative water content between the three drought stress levels are significantly different, with the average of 74.8~85.3% for the well-watered control group, 38.3~45.2% for the

**Table 3. Soil relative water content.**

| Plant species | Well-watered group (%) | | Moderate stress group (%) | | Severe stress group (%) | |
|---|---|---|---|---|---|---|
| | 27th, April 2015* | 29th, April 2015** | 2ed, May 2015* | 4th, May 2015** | 7th, May 2015* | 9th, May 2015** |
| *P. lactiflora* | 80.1±2% | 75.8±1% | 45.2±4% | 42.1±2% | 20.3±2% | 18.2±2% |
| *He. dumortieri* | 82.3±4% | 76.8±2% | 41.6±2% | 40.7±1% | 17.6±4% | 15.5±3% |
| *M. urticifolia* | 78.8±5% | 75.2±1% | 41.9±3% | 40.5±2% | 17.2±2% | 15.4±2% |
| *I. lactea* | 85.3±1% | 80.2±3% | 42.5±3% | 41.3±3% | 21.3±3% | 19.7±1% |
| *He. dumortieri* | 82.9±2% | 79.3±2% | 40.8±3% | 49.8±4% | 21.1±2% | 18.6±2% |
| *S. lineare* | 78.4±1% | 76.1±2% | 43.4±1% | 40.2±3% | 24.3±3% | 22.4±3% |
| *I. germanica* | 85.3±2% | 78.5±2% | 44.7±2% | 41.4±3% | 22.4±2% | 20.2±3% |
| *C. bipinnata* | 77.1±2% | 74.8±3% | 40.1±3% | 39.2±2% | 20.7±4% | 17.8±4% |
| *Ho. plantaginea* | 83.8±3% | 78.6±2% | 42.8±2% | 38.3±2% | 23.4±3% | 22.3±5% |
| *D. barbatus* | 82.3±4% | 79.0±4% | 40.3±1% | 39.6±2% | 24.3±1% | 22.5±2% |

The values represent the mean ± SD ($n = 30$).

* is the day when the water content of the soil achieved the designated level.

** is the sampling day.

moderate drought level, and 15.4~24.3% for the severe drought level (Table 3). These findings were consistent with the designed levels.

## 4.2 Membrane permeability

Plant cells dehydrate when they suffer drought stress, which leads to mechanical damage to the membranes [72]. Greater MP values led to more cytosolic exosmosis and further damage to the plant cellular structure. However, it was hard to distinguish which species had stronger or weaker drought resistance when they were under the well-watered control because they had not been affected by drought yet. In this study, *C. bipinnata* and *D. barbatus* presented significantly higher MP values than the other species (Fig 1), followed by the MP value of *M. urticifolia* and *Hy. Erythrostictum*, for which the MP value was significantly different relative to the remaining species. On the contrary, Liliaceae and Iridaceae family species presented low MP values under the well-watered control. The change of MP values implies the different physiological characteristics of various plants.

With the drought stress treatment, the above four plant species with higher MP values expressed different tolerance features. *C. bipinnata* did not survive under severe drought stress, and *M. urticifolia* did not survive under moderate or severe drought stress. Both species are annual herbs, and their root growth is strongly inhibited by the lack of water [73,74]. Although leaf sampling occurred only ten days after transplanting, the plant roots were transplanted with the original moist rooting medium and the seedlings were shaded and fully watered, which was beneficial for root development. Previous studies identified a strong relationship between new leaf germination and plant survival rate [75]. In addition, leaf biomass was positively correlated with root biomass, implying that root length developed when the plants grew new leaves [76]. In this study, plants under the well-watered control have all grown new leaves and even buds at sampling, thus demonstrating that the root has developed. Leaf sampling could be conducted in 3 days, 5 days, 10 days, etc. once the soil RWC matched the designed drought levels [77–79]. Therefore, the withering of *C. bipinnata* and *M. urticifolia* was induced by drought stress instead of a short growth period. The MP values of *D. barbatus* did not change significantly.

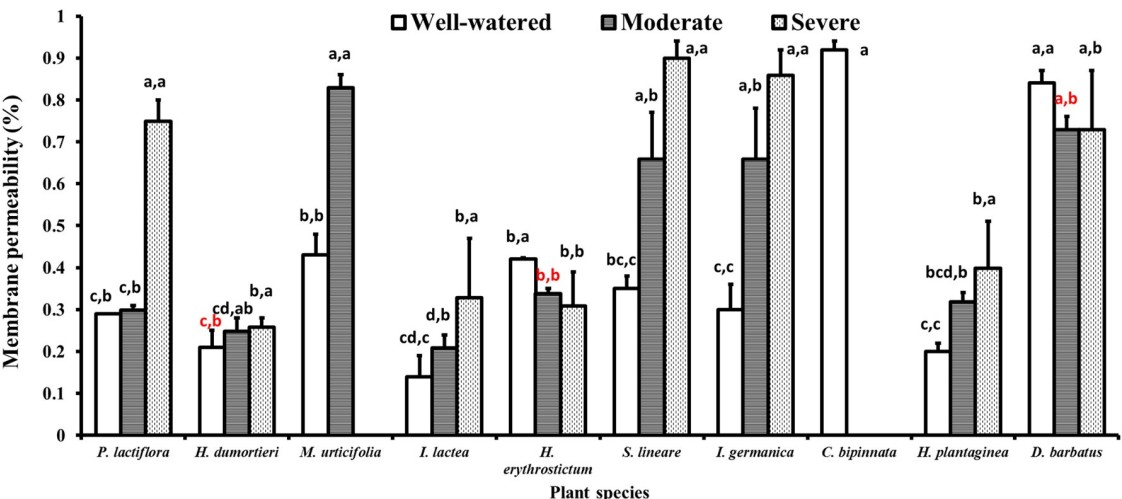

**Fig 1. Variance of the membrane permeability for the ten species under drought stress.** The histogram shows the mean values. Above the histogram, the lowercase letters before the commas indicate statistical significance among the different plant species, and those after the commas indicate statistical significance among the well-watered, moderate and severe drought stress treatments. The different lowercase letters indicate a significant difference at $P<0.05$.

Under drought stress, most species showed significantly increased MP values, including *P. lactiflora*, *He. dumortieri*, *I. lactea*, *S. lineare*, *I. germanica*, and *Ho. plantaginea*, among of which *P. lactiflora* showed significantly increased MP values only under severe drought stress (Fig 1). The MP value of *I. germanica* increased the most by 36% and 56% under moderate and severe conditions, respectively, and that of *S. lineare* increased by 31% and 55%, respectively. These findings indicated that these species' membranes were damaged under drought stress.

Among all the plant species, only two, *Hy. erythrostictum* and *D. barbatus*, did not show significantly changed MP values under the drought stress treatment. *Hy. erythrostictum* was considered a kind of xerophilous plant with fleshy leaves [80], and the MP values were reduced at the early stage of drought stress [81]. Although fewer investigations have been performed on the effects of drought in *D. barbatus*, especially on its MP, the permeation regulation synchronized with damage to the protoplast membrane in *Dianthus plumarius*, another Caryophyllaceae plant [82]. Although both *S. lineare* and *Hy. erythrostictum* belong to the Crassulaceae family, their MP value variation trend was the opposite. The MP values of *S. lineare* were also significantly greater than that of *Hy. erythrostictum* under drought stress [83].

Plant drought resistance is closely related to its cell membrane system stability [31]. Usually, the cell membrane is first affected by drought stress [84], and then the cell structure is damaged and MP increases, which leads to the extravasation of extracellular electrolytes, which is why MP values increase when plants are subjected to drought stress. The stability of the MP values of these two species indicated that their cell membrane was undamaged. Therefore, the osmotic adjustment ability of multicolored carnation leaves is strong enough to avoid damage to the protoplast membrane under drought stress treatment.

### 4.3 Chlorophyll concentration

In the well-watered control, the studied plant species presented Chl concentrations from 6.06 to 47.69 mg·g$^{-1}$ FW, and the fleshy Crassulaceae species *H. erythrosticum* and *S. lineare* presented the lowest Chl concentrations (Fig 2). The plants' Chl concentrations were affected by light intensity and environmental temperature, which affect the opening and closing of

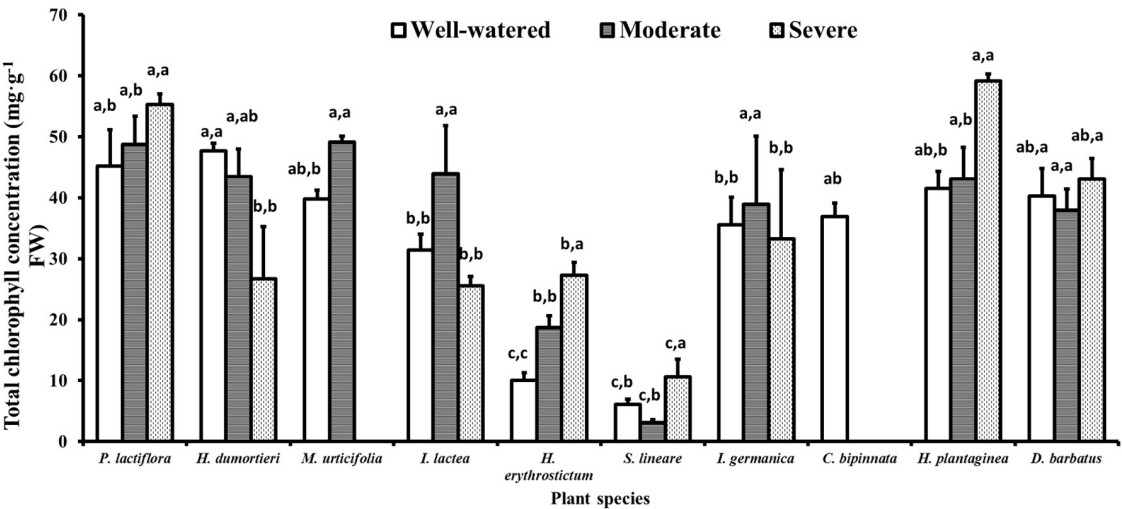

**Fig 2. Variance of the total chlorophyll concentration in the ten species during the drought stress tests.** The histogram shows the mean values. Above the histogram, the lowercase letters before the commas indicate statistical significance among the different plant species, and the lowercase letters after the commas indicate statistical significance among the well-watered, moderate and severe drought stress treatments. The different lowercase letters indicate a significant difference at $P<0.05$.

stomata and photosynthetic rates of plant leaves and then affect the accumulation of carbohydrates, which is consistent with the Chl concentration of plants [85]. In the study, all plants grew in a stable light intensity environment because the experimental area had a roof, which can prevent the effects of strong sunshine or rain. As for the ambient temperature, no extreme temperatures were encountered during the experiment. The leaves of these plants were collected at the same time after drought stress under the same environmental temperature. Thus, the changes in Chl concentration should be caused by drought stress instead of temperature or sunlight.

Upon drought stress, change trends of plant Chl concentration were various. Four species showed significantly ($P<0.05$) increased Chl concentrations under the different extents of drought stress, including *P. lactiflora*, *Hy. erythrostictum*, *S. lineare* and *Ho. plantaginea* (Fig 2). The increased Chl concentration under moderate or severe drought stress might be due to the increase of the stem cell mass and cell number of the leaves, thus forming a Chl condensation phenomenon, as in *P. lactiflora* [86]. Additionally, Liu *et al* [87] reported that the Chl a and b of *Hy. erythrostictum* would be increased during the day but decreased during the night under drought stress, which probably indicates more photosynthetic pigments were produced to promote photosynthesis of *Hy. erythrostictum* under drought stress. However, the photosynthetic pigment content was decreased at night for maintaining its normal physiological activities.

The species *I. germanica* and *I. lacteal* showed increased Chl concentrations under moderate stress, and then these values decreased under severe drought stress (Fig 2). Most Iridaceae plants are shade plants [88], some of which feature colorless leaves [89] and possess lower Chl concentrations than sun plants leaves [90]. Zhou [91] researched seven Iris species and also found that *I. germanica* had a higher Chl concentration in the early stage of drought stress than the control group.

*D. barbatus* did not change the Chl concentration under drought stress treatment (Fig 2), which indicated that the species had a strong self-repair and regulate ability during drought

stress, and its leaves had a relatively good physiological and biochemical state, which could maintain normal photosynthesis and strong resistance during drought.

Only one species, *He. Dumortieri*, significantly decreased the Chl concentration with the increase of drought stress (Fig 2), which indicated that chlorophyll synthesis was interrupted and the chlorophyll decomposed under drought.

## 4.4 Superoxide dismutase activity

SOD activity is very sensitive to drought [92]. In this study, the SOD activity of six species, *He. dumortieri*, *I. lactea*, *Hy. erythrostictum*, *I. germanica*, *Ho. plantaginea*, and *D. barbatus*, initially significantly ($P<0.05$) decreased under moderate drought stress but increased under severe drought stress (Fig 3). The reduction of SOD activity under the moderate condition implied that a considerable amount of ROS was produced to damage plant cells and tissues, thus leading the plant cells to undergo oxidative damage. The activity of the enzyme SOD is influenced by the concentration of the $O_2^-$ substrate. Stress raises the production of $O_2^-$, thus increasing the SOD activity [93]. A previous investigation indicated that the increased SOD activity under severe drought was caused by the drought exercise under moderate drought stress [87]. The drought exercise was applied to enhance the resistance of rice to high temperatures [94] and the resistance of wheat to drought stress [95].

The change trends of SOD activity can be adopted to judge the species' drought resistance. Those species with higher SOD activity under drought can be considered to have strong drought resistance [96]. The SOD activity of *P. lactiflora* and *S. lineare* increased under either the moderate or severe drought stress. The difference of SOD activity might be caused by the expression of various isozymes, which induced the accumulation of the antioxidant substance in plants leaves that started up the antioxidant protection system when plants were under moderate and severe drought stress [97]. The degrees of SOD increase of *S. lineare* were higher than that of *P. lactiflora*, which is consistent with a previous study [98] in which *S. lineare* had stronger SOD activity.

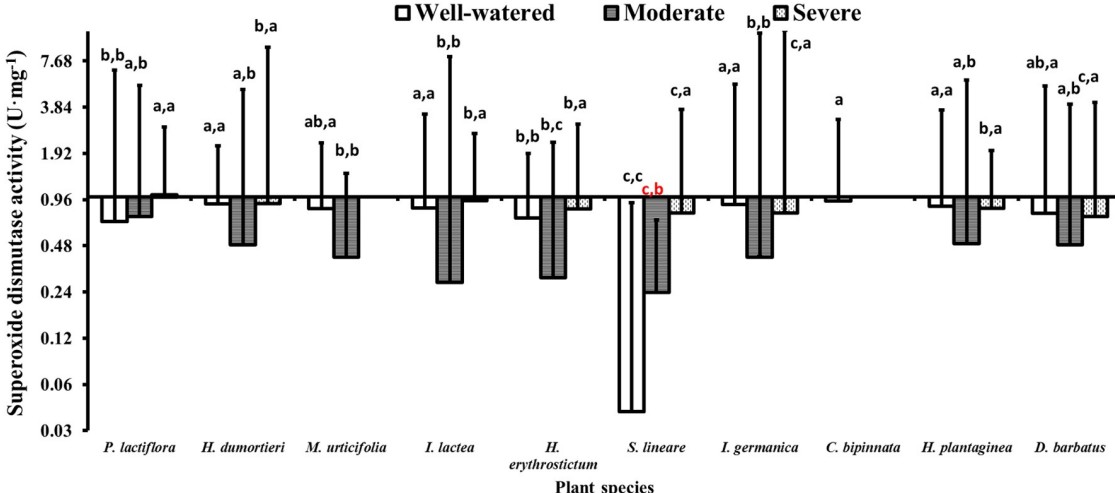

**Fig 3. Variance of superoxide dismutase activity in the ten species during the drought stress tests.** The histogram shows the mean values. Above the histogram, the lowercase letters before the commas indicate statistical significance among the different plant species, and the lowercase letters after the commas indicate statistical significance among the well-watered, moderate and severe drought stress treatments. The different lowercase letters indicate a significant difference at $P< 0.05$.

Overall, in this study, the SOD activity in most species increased after severe drought stress, which suggested that drought stress-induced SOD activity increases in these plant species to help them eliminate ROS. These plant species presenting increased SOD activity showed advantages in terms of their drought stress response, and proper drought exposure could significantly improve plant resistance to sustained drought stress [95].

## 4.5 Peroxidase activity

In the study, all plant species showed decreased POD activity under the moderate or severe drought stress, with most showing declining POD activity as the drought stress increased (Fig 4). Compared with the well-watered control, the POD activity of *He. dumortieri*, *Hy. erythrostictum*, *S. lineare*, *I. germanica*, *Ho. plantaginea* and *D. barbatus* was reduced significantly ($P<0.05$) during the moderate drought stress. The POD activity of these species was significantly ($P<0.05$) reduced under the severe drought stress (Fig 4). Such declining trends of POD activity with drought stress were contrary to the increased SOD activity trends.

Under drought stress, stronger POD activity might be attributed to the plant defense mechanisms against free radical formation resulting from water deficit [99]. According to Fig 4, the POD activity of *I. lacteal* was reduced under both the moderate and severe drought stress and was significantly ($P<0.05$) higher than that of the species that survived under severe drought stress except for *He. dumortieri* and *Ho. plantaginea*. This finding may indicate the species that have stronger resistance to drought.

Under severe drought stress, *P. lactiflora*, *S. lineare*, *I. germanica* and *D. barbatu* showed significantly lower POD activity values compared with the other species and other water conditions (Fig 4), which may be related to the POD enzyme reaching its tolerance limit and decreasing rapidly.

In general, the POD activity of all the species tended to decrease when the seedlings experienced drought stress. As a special enzyme to eliminate $H_2O_2$, the reduced POD activity in this study might have been caused by ROS elimination because different protective enzymes work together as a whole, with the elimination of $O_2^-$ SOD increasing $H_2O_2$ production. However, a very high concentration of $H_2O_2$ was beyond the reach of POD activity [100],

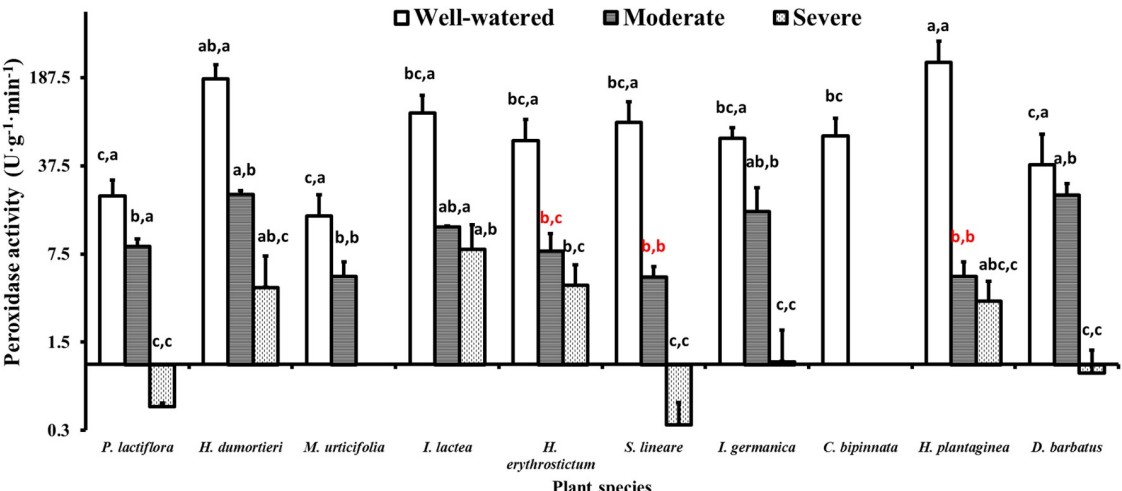

**Fig 4. Variance of peroxidase activity of the ten species under the drought stress tests.** The histogram shows the mean values. Above the histogram, the lowercase letters before the commas indicate statistical significance among the different plant species, and the lowercase letters after the commas indicate statistical significance among the well-watered, moderate and severe drought stress treatments. The different lowercase letters indicate a significant difference at $P<0.05$.

which indicates that although SOD and POD are both antioxidases and can cooperate in scavenging ROS, drought stress might lead to a different enzyme system to resist adverse drought environments.

## 4.6 Ascorbate peroxidase activity

Under well-watered conditions, *P. lactiflora* presented the highest APX activity value at 0.98 U·min$^{-1}$·g$^{-1}$ FW, although the other species had APX activity from 0.09 to 0.26 U·min$^{-1}$·g$^{-1}$ FW. Seven species showed increased APX activity, with *P. lactiflora*, *He. dumortieri*, *Hy. erythrostictum*, *I. germanica*, *Ho. plantaginea*, and *D. barbatus* presenting significantly higher APX values under moderate drought than under well-watered and severe drought stress conditions ($P < 0.05$). However, the APX activity of *S. lineare* decreased significantly ($P < 0.05$) under moderate and severe drought stress. At severe stress, most plant species reduced their APX value no more than 0.50 U·min$^{-1}$·g$^{-1}$ FW, with the lowest at 0.07 U·min$^{-1}$·g$^{-1}$ FW for *Ho. plantaginea* (Fig 5).

An interesting phenomenon in this study was that the SOD and APX activity of some plants seemed to complement each other. The SOD activity of *He. dumortieri*, *I. lactea*, *Hy. erythrostictum*, *I. germanica*, *Ho. plantaginea*, and *D. barbatus* decreased under moderate drought stress and increased under severe stress (Fig 3), whereas the APX activity displayed the opposite pattern. Moreover, the highest values of APX activity in these species were recorded in the moderate drought stress treatment, which may suggest that APX activity was first activated by the early or moderate drought stress to scavenge ROS. When the SOD activity increased under severe drought stress, the APX activity decreased at the same time, which suggests that SOD plays a dominant role in ROS scavenging during severe drought stress. The weakening of APX activity under severe drought stress indicates that its antioxidant capability is temporary and limited [101].

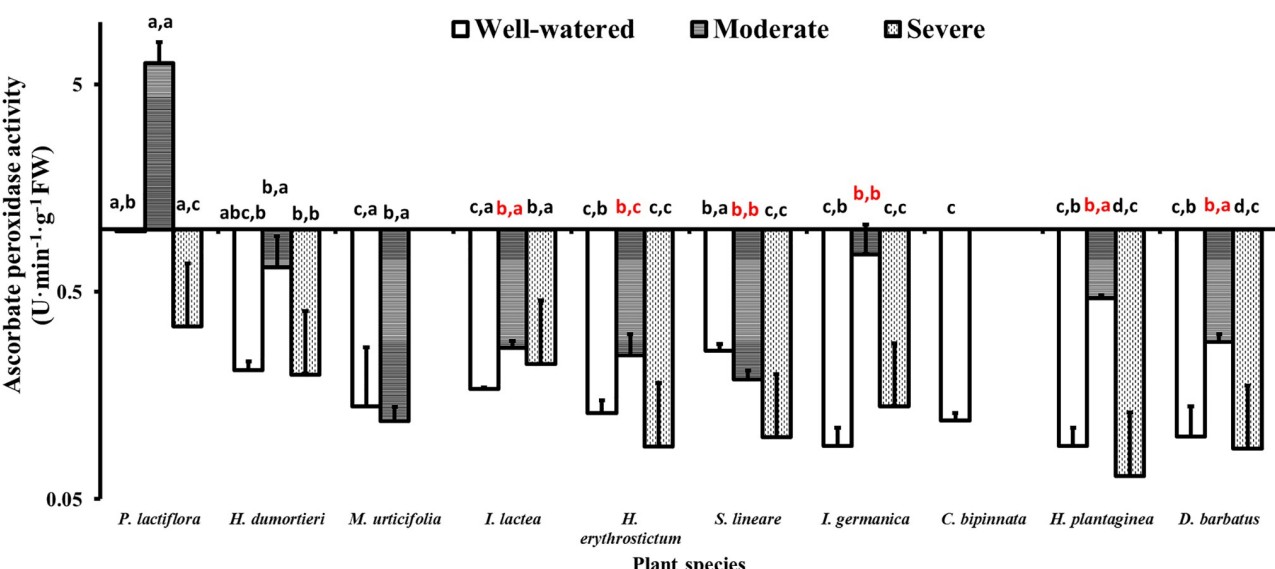

**Fig 5. Variance of ascorbate peroxidase activity of the ten species under the drought stress tests.** The histogram shows the mean values. Above the histogram, the lowercase letters before the commas indicate statistical significance among the different plant species, and the lowercase letters after the commas indicate statistical significance among the well-watered, moderate and severe drought stress treatments. The different lowercase letters indicate a significant difference at $P < 0.05$.

**Table 4. Drought resistance under the drought stress test.**

| Plant species | MP | Chl | SOD | POD | APX | $\overline{X}_i$ | Drought resistance rank |
|---|---|---|---|---|---|---|---|
| P. lactiflora | 0.34 | 0.45 | 0.39 | 0.46 | 0.37 | 0.4021 | 8 |
| He. dumortieri | 0.60 | 0.60 | 0.66 | 0.37 | 0.34 | 0.5140 | 1 |
| M. urticifolia | - | - | - | - | - | Non available | 9 |
| I. lactea | 0.46 | 0.44 | 0.62 | 0.35 | 0.52 | 0.4772 | 3 |
| Hy. erythrostictum | 0.42 | 0.50 | 0.60 | 0.36 | 0.42 | 0.4608 | 5 |
| S. lineare | 0.52 | 0.46 | 0.42 | 0.35 | 0.52 | 0.4542 | 6 |
| I. germanica | 0.55 | 0.47 | 0.60 | 0.42 | 0.36 | 0.4795 | 2 |
| C. bipinnata | - | - | - | - | - | Non available | 10 |
| Ho. plantaginea | 0.53 | 0.36 | 0.65 | 0.34 | 0.35 | 0.4460 | 7 |
| D. barbatus | 0.33 | 0.49 | 0.63 | 0.52 | 0.35 | 0.4640 | 4 |

$\overline{X}_i$ refers to the Membership function value.

Higher $\overline{X}_i$ value is, stronger plant drought resistance is.

## 4.7 Assessment of plant drought resistance

We determined the drought resistance of ten ground cover species through six physiological indicators: MP, Chl, SOD activity, POD activity, and APX activity. However, it is difficult to judge which plant species has better drought resistance only based on individual parameters. Therefore, it is reasonable to use the membership function method that applies fuzzy mathematics to weigh these indicators and ultimately assess the drought resistance of the ten species. The calculated results following the formula of the membership function are shown in Table 4. The findings indicate that *He. dumortieri* was the most drought resistant species while *C. bipinnata* and *M. urticifolia* were not suitable for moderate or severe drought stress due to withering. In addition, *P. lactiflora* survived the weakest drought resistance. The order of plant resistance to drought stress was as follows: *He. dumortieri* > *I. germanica* > *I. lactea* > *D. barbatus* > *Hy. erythrostictum* > *S. lineare* > *Ho. plantaginea* > *P. lactiflora* > *M. urticifolia* > *C. bipinnata* (Table 4).

## 5 Conclusions

This study investigated how ten common plant species were tolerant to levels of drought stress and showed that drought stress disrupted plant growth because the same conditions were not observed under the well-watered treatment. Five parameters (MP, Chl, SOD, POD, and APX activity) changed under moderate and severe drought stress. The main results are as follows.

First, *C. bipinnata* and *M. urticifolia* failed to survive the drought stress and were not suitable for both moderate and severe drought stress.

Second, each plant species had quite different physiological and biochemical parameters. *He. dumortieri*, *I. lactea*, and *Ho. plantaginea* maintained a stable MP value after experiencing drought stress. Most species (except *P. lactiflora* and *S. lineare*) showed reduced SOD activity under moderate drought stress but increased activity under severe drought stress. However, the plant species showed decreased POD activity and APX activity when the drought stresses increased.

Third, complementary relationships might occur among SOD, POD and APX activity, and SOD may play a dominant role in scavenging ROS under severe drought stress while APX and POD are responsible under moderate drought stress.

Finally, *C. bipinnata* and *M. urticifolia* were very sensitive to drought stress and thus are unfit for roof greening, especially in arid regions. However, *He. dumortieri*, *I. germanica*, *I. lactea*, *D. barbatus*, *Hy. erythrostictum*, *S. lineare*, *Ho. plantaginea*, and *P. lactiflora* could be applied as roof greening in Beijing and other northern Chinese cities.

## Supporting information

**S1 Data.**
(XLS)

## Acknowledgments

The authors thank the anonymous reviewers and editor Saddam Hussain for valuable comments, and Fen Qin (a business development manager for the China office from Power Systems Research) for helping with the manuscript preparation.

## Author Contributions

**Conceptualization:** Pengqian Zhang, Jiade Bai, Yanju Liu.

**Data curation:** Pengqian Zhang.

**Formal analysis:** Pengqian Zhang, Jiade Bai.

**Funding acquisition:** Pengqian Zhang, Yanju Liu.

**Investigation:** Pengqian Zhang, Zheng Yang, Tian Liu.

**Methodology:** Pengqian Zhang, Yanju Liu, Zheng Yang, Tian Liu.

**Project administration:** Pengqian Zhang, Jiade Bai, Yuping Meng.

**Validation:** Yanju Liu.

**Writing – original draft:** Pengqian Zhang.

**Writing – review & editing:** Yanju Liu.

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
