## [Editor Report · Decision Letter 0]

20 Aug 2019

PONE-D-19-20236

Drought Resistance of 10 Ground Cover Seedling Species during Roof Greening

PLOS ONE

Dear Dr. Liu,

Thank you for submitting your manuscript to PLOS ONE. After careful consideration, we feel that it has merit but does not fully meet PLOS ONE’s publication criteria as it currently stands. Therefore, we invite you to submit a revised version of the manuscript that addresses the points raised during the review process.

PLOS ONE requires that all submitted research be written in clear, correct, unambiguous English (https://journals.plos.org/plosone/s/criteria-for-publication#loc-5). In this case, we feel that your manuscript does not meet this criterion. At this time, we would therefore ask that you have your manuscript thoroughly copyedited for spelling and grammar.

We would appreciate receiving your revised manuscript by Oct 04 2019 11:59PM. To enhance the reproducibility of your results, we recommend that if applicable you deposit your laboratory protocols in protocols.io, where a protocol can be assigned its own identifier (DOI) such that it can be cited independently in the future. For instructions see: http://journals.plos.org/plosone/s/submission-guidelines#loc-laboratory-protocols

We look forward to receiving your revised manuscript.

Kind regards,

Dr Jamie Males

Associate Editor

PLOS ONE

on behalf of

Dorin Gupta, Ph.D.

Academic Editor

PLOS ONE
---

## [Author Response · Author response to Decision Letter 0]

18 Sep 2019

Dear, editor, 

I appreciate your attention to the manuscript.

We have edited the manuscript very carefully in this version:

Firstly, we revised the grammar and spelling of the manuscript completely that included tense, voice, syntactical structure, and some spelling mistakes.

Secondly, we deleted some words that had nothing to do with the manuscript itself.

Kind regards,

Dr. Yanju-Liu

---

## [Decision Letter · Decision Letter 1]

7 Jan 2020

PONE-D-19-20236R1

Drought Resistance of 10 Ground Cover Seedling Species during Roof Greening

PLOS ONE

Dear Dr. Liu,

Thank you for submitting your manuscript to PLOS ONE. After careful consideration, we feel that it has merit but does not fully meet PLOS ONE’s publication criteria as it currently stands. Therefore, we invite you to submit a revised version of the manuscript that addresses the points raised during the review process.

We would appreciate receiving your revised manuscript by Feb 21 2020 11:59PM. To enhance the reproducibility of your results, we recommend that if applicable you deposit your laboratory protocols in protocols.io, where a protocol can be assigned its own identifier (DOI) such that it can be cited independently in the future. For instructions see: http://journals.plos.org/plosone/s/submission-guidelines#loc-laboratory-protocols

We look forward to receiving your revised manuscript.

Kind regards,

Saddam Hussain

Academic Editor

PLOS ONE

Additional Editor Comments (if provided):

After careful consideration of myself and based on the evaluation of external reviewers, I feel that the manuscript is interesting, but is not suitable for publication as it currently stands. A "Major Revision" is required as the comments of reviewers. I am in the view of reviewer 1, that the authors provided a large set of data, but failed to provide any justification or explanation of results, or why they found inconsistent patterns in the data. Results and Discussion section should be thoroughly improved. Present your key findings in the conclusion section. Change the description of treatments as well-watered’, ‘moderate’ and ‘severe’ drought. Title should also be improved. Replace number (10) with text (ten). Authors should also carefully improve the language of the draft, to omit the minor grammatical and typo mistakes.

Reviewers' comments:

Reviewer's Responses to Questions

**Comments to the Author**

1. If the authors have adequately addressed your comments raised in a previous round of review and you feel that this manuscript is now acceptable for publication, you may indicate that here to bypass the “Comments to the Author” section, enter your conflict of interest statement in the “Confidential to Editor” section, and submit your "Accept" recommendation.

Reviewer #1: (No Response)

Reviewer #2: All comments have been addressed

2. Is the manuscript technically sound, and do the data support the conclusions?

Reviewer #1: Partly

Reviewer #2: Yes

3. Has the statistical analysis been performed appropriately and rigorously? 

Reviewer #1: Yes

Reviewer #2: Yes

4. Have the authors made all data underlying the findings in their manuscript fully available?

Reviewer #1: Yes

Reviewer #2: Yes

5. Is the manuscript presented in an intelligible fashion and written in standard English?

Reviewer #1: Yes

Reviewer #2: Yes

6. Review Comments to the Author

Reviewer #1: This paper assesses the suitability of 10 species for use on green roofs according to their drought tolerance. Drought tolerance was quantified using a number of enzymes extracted from leaves after exposure to three treatments modifying water availability. A composite drought tolerance index was then used to rank species according to their suitability for use on green roofs.

Overall, the paper is well-written, and the methods are largely sound. I find, however, that the results and discussion do not do a good job of putting the results in the context of the literature. What I mean is, for each response variable measured (i.e., for each subsection of the results/discussion), a paragraph is first given which describes how to effectively interpret the response variable and what conditions affect the variable in question. Take Section 4.2 for example. The authors explain what a high/low value of MP means and what can affect it --- in very general terms. Then, the remaining paragraphs are equivalent to just ‘results’ --- that is, the next three paragraphs describe the trends and statistical differences among species in relation to watering treatment, but do not attempt to explain or discuss what they have observed in the context of literature. Further, the implications of the findings are not expressed, and no conclusions are presented. All we have is a description of the results, but no further explanation. For example --- why have certain species been less affected by the drought with regard to membrane permeability? What do we know about these species, their adaptations, where they come from --- that is, are the results you found logical? As the reader, we cannot assess whether your findings are sensible, nor can we walk away from this paper with anything except species rankings, without an explanation of the results. The MP results are clear, compared with SOD. Looking at Figure 3, we can see for most species SOD is high for Cg, low in T1, then high again in T2 --- why is this? As it stands, the species are ranked only within treatment (i.e., all species exposed to Cg are ranked, then ranked within T1, then ranked within T2 --- so three different rankings are produced only within each treatment). What we need to see is your explanation as to why we did not see a clear increase or decrease in SOD in relation to declining water ability --- i.e., the comparison among treatments, within each species. I cannot accept the rankings as they are, because there is no clear trend in SOD in relation to treatment and more importantly, no attempt is made to explain why we see the results as presented in Figure 3. Section 4.6 has great potential to present a synthesis of why the variables contributing to ‘Xi’ respond to watering treatment in the way do --- that is, once the trends in each response variable have been explained in isolation, a more sophisticated discussion of the combined response of all variables is required --- such that we can understand the overall strategy of each species and therefore present a justified species ranking --- that is, a ranking supported by what we know and understand about these species, such that the ranking as presented according to the synthesised variable Xi can be properly assessed. As it is now, we must take your word for it that this ranking is sensible. In my view, this is the major weakness of the paper, as all sections in the results/discussion fail to properly interrogate and explain the trends in the data.

Some criticism of the length of time plants were exposed to their treatments prior to measurement is likely. Plants were only exposed to their treatment soil water content for two days, which is quite short. However, differences in some response variables were observed, however, they were inconsistent at times which may be due to the way drought was imposed. Further justification of the suitability of this type of drought exposure would help. I can see the average soil moisture content values in Table 3, which I presume is the content for the two days of exposure. However, the treatments began when the watering regime was imposed, so I suggest a figure is required to show the decline in soil moisture content over time, to get an understanding of the severity of drought treatments over time, not just in the two days prior to sampling. I am not familiar with how long it takes for the five measured parameters to change in response to drought, so discussion of this is required.

I also suggest that the use of ‘Cg’, ‘T1’ ‘T2’ and are not intuitive labels when interpreting the results and would suggest re-labelling them as something like: ‘well-watered’, ‘moderate’ and ‘severe’ drought treatments which makes it easier for the reader. Similarly, codes for species should be used on the x-axis, as numbering species is not intuitive and makes the reader work hard to find out which species is what. This should be done in the text of the results/discussion, as well as on all tables and figures. Further, log scales on figures such as Figures 4 and 5 may improve their readability, rather than using broken y-axes.

Reviewer #2: Title: Drought Resistance of 10 Ground Cover Seedling Species during Roof Greening from Zhang et al. which was submitted to PloS One has meaningful results and well conducted experiment. The use of different plant species i.e. Paeonia lactiflora, Hemerocallis dumortieri, Physostegia virginiana, Iris lacteal, Hylotelephium erythrostictum, Sedum lineare, Iris germanica, Cosmos bipinnata, Hosta plantaginea and Dianthus barbatus for determining the drought tolerant and drought sensitive species is a novel research. I recommend this article should be publish in PloS One before changing some minor mistakes which are as follow:

On the abstract: You have mentioned Cosmos bipinnata and Physostegia virginiana died while other plant species survived. I recommend you please change the terminology like died is not suitable for the plant you can write in a better way.

On the introduction: The information regarding ROS or oxidative stress is too limited. You should add at least 2-3 sentences about their mechanism, effect and actions of antioxidants in response to oxidative stress. https://doi.org/10.1016/j.ecoenv.2019.109915. doi:10.3390/plants8120545. https://doi.org/10.1007/s11356-019-07264-7. Please see these latest articles which just published recently and find some relevant information about your topic and cites these in introduction.

On Material and Method: You have mentioned “The seedlings were provided by the Yu-quanying flower market, a large and popular market in Beijing” can you provide more details about it (if possible).

You have mentioned “mg•kg-1” Please see above mention articles and see how we can correctly write this unit and write correctly throughout your manuscript.

You have mentioned “Leaves were placed into sealed plastic bags and kept in a portable ice box at 0 ~ 4℃” Scientifically when you are collecting your samples please took in liquid nitrogen. You can taken in the ice box. It’s ok but more convenient to take in liquid nitrogen.

On Results and discussions: Better to write chl a like this: use first time full abbreviation.

“Drought stress causes changes in chlorophyll content in plants” Please modify this sentence.

You have find that “The chlorophyll content of Physostegia virginiana was the highest, at 49.07 mg/g•FW” This is the maximum chlorophyll contents and determined by Arnon’s (1949) method. Are you sure that you determined too much chlorophyll in this plant species. For the best of our knowledge we did not saw too much high chlorophyll contents. Please verify your formula or your method and make sure that chlorophyll is very high? Please see it carefully.

“The chlorophyll content of Hosta plantaginea was the highest, at 59.11 mg/g•FW” Is it?

Please verifily the units of antioxidants you have determined “U•min-1 •g -1 FW” or “U•mg-1” Please make sure that you are going through with right units.

7. PLOS authors have the option to publish the peer review history of their article (what does this mean?). If published, this will include your full peer review and any attached files.

Reviewer #1: No

Reviewer #2: No

---

## [Author Response · Author response to Decision Letter 1]

25 Feb 2020

Response to Reviewers

First, we are very grateful to the reviewers for their careful work and useful advice, which are truly beneficial for improving the quality of the article. We have accepted the comments of the reviewers and editor and have tried our best to revise the article as needed.

Response to Reviewer #1’s main comments:

Summary comment: The section of results and discussion required a “Major Revision”. 

Summary response: Yes. Thanks for review’s nice suggestions. We have revised the manuscript according to each of the comments as follows.

Comments 1: The explain of MP, Chl, SOD, POD, and AsAPOD(APX) means required greater depth.

Response 1: We looked up a large number of papers and cited 31 of them to elaborate further on the importance of the MP, Chl, SOD activity, POD activity, and APX activity indicators. Not only was the first paragraph of each section (section 4.2 to 4.6) supplemented with explanations of the indicators but also some scientific discussions were interspersed throughout other paragraphs.

Comments 2: The describes of statistical differences among species is not enough, it needs further explain why species have such change under the watering condition.

Response 2: We have strengthened the discussion of each indicator. On the basis of the objective elaboration of the statistical data, an interpretation of the changes in the data has been added. 

For instance, in section 4.2, we further explained the cause of death of Cosmos bipinnata and M. urticifolia under drought stress, explained the reason why C. bipinnata presented increased MP values and further explained the why the stability of the various membrane systems of the cells means strong resistance to drought stress. 

In section 4.3, we first summarized the general trends of plant Chl before the detailed discussion. We added a paragraph to discuss the changes in different species from the family perspective, which made the discussion are more reasonable. 

Similarly, we have made in-depth revisions to section 4.4. We summarized three suggestions: (1) drought stress induced an increase in SOD activity, and by increasing their SOD activity, the abovementioned plant species eliminated ROS; (2) the plant species presenting increased SOD activity showed advantages in terms of their drought stress response; and (3) proper drought exposure could significantly improve plant resistance to sustained drought stress.

We also strengthened the link among the POD, APX, and SOD protective enzymes in the following two sections. We found that there may be a complementary relationship among POD, APX, SOD. On the basis of the change trends of the activity of these three protective enzymes, SOD may play a dominant role in scavenging ROS during severe drought stress (or during the late period of drought stress), and APX and POD may play a dominant role in scavenging ROS during moderate drought stress (or during the early period of drought stress).

In accordance with the reviewer's suggestion, we added details to Table 3 (the moisture content of the soil and the increases in soil RWC on the sampling day).

Comments 3: Figures 4 and 5 may improve their readability, rather than using broken y-axes.

Response 3: We improved the figures in the paper. At first, “Cg”, “T1” and “T2” were used instead of “well-watered”, “moderate” and “severe”, respectively. Secondly, Figures 4 and 5 have been changed into log-scale figures.

Response to Reviewer #2’s comments:

Comments 1: On the abstract: You have mentioned Cosmos bipinnata and Physostegia virginiana died while other plant species survived. I recommend you please change the terminology like died is not suitable for the plant you can write in a better way.

Response 1: We used the word “is not suitable for” instead of “died”.

Comments 2: On the introduction: The information regarding ROS or oxidative stress is too limited. You should add at least 2-3 sentences about their mechanism, effect and actions of antioxidants in response to oxidative stress.

Response 2: Following reviewer’s suggestion, we not only supplemented some descriptions of ROS but also added many descriptions to other parts of the discussion.

Comments 3: On Material and Method: You have mentioned “The seedlings were provided by the Yu-quanying flower market, a large and popular market in Beijing” can you provide more details about it (if possible).

Response 3: We have added some a specific description of the market.

Comments 4: You have mentioned “mg•kg-1” Please see above mention articles and see how we can correctly write this unit and write correctly throughout your manuscript.

Response 4: We checked the reference of “ZHU Minghao, et al. The impact of Elaphurus davidianus in different habitats on soil physical and chemical properties [J]. Environmental Chemistry, 2016, 35(1): 208-217. DOI:10.7524/j.issn.0254-6108.2016.01.2015070803”. Again, units of “mg•kg-1” were correct.

Comments 5: Better to write chl a like this: use first time full abbreviation.

Response 5: We have replaced “Chl a” by “Chlorophyll a” when it appeared first time.

---

## [Decision Letter · Decision Letter 2]

24 Mar 2020

PONE-D-19-20236R2

Drought Resistance of 10 Ground Cover Seedling Species during Roof Greening

PLOS ONE

Dear Dr. Liu,

Thank you for submitting your manuscript to PLOS ONE. After careful consideration, we feel that it has merit but does not fully meet PLOS ONE’s publication criteria as it currently stands. Therefore, we invite you to submit a revised version of the manuscript that addresses the points raised during the review process.

ACADEMIC EDITOR: Although, the authors tried to improve the manuscript compared with first draft, but failed to respond/deal with the comments raised by me (editor) and reviewer 1. A major revision and detailed point by point response is required prior to publication. Also respond to the editor's comments raised during revision 1.

Authors should give proper explanation of results, or why they found inconsistent patterns in the data. Results and Discussion section should be thoroughly improved. Present your key findings in the conclusion section. Change the description of treatments as well-watered’, ‘moderate’ and ‘severe’ drought. Title should also be improved. Replace number (10) with text (ten). Authors should also carefully improve the language of the draft, to omit the minor grammatical and typo mistakes.

We would appreciate receiving your revised manuscript by May 08 2020 11:59PM. To enhance the reproducibility of your results, we recommend that if applicable you deposit your laboratory protocols in protocols.io, where a protocol can be assigned its own identifier (DOI) such that it can be cited independently in the future. For instructions see: http://journals.plos.org/plosone/s/submission-guidelines#loc-laboratory-protocols

We look forward to receiving your revised manuscript.

Kind regards,

Saddam Hussain

Academic Editor

PLOS ONE

Additional Editor Comments (if provided):

Although, the authors tried to improve the manuscript compared with first draft, but failed to respond/deal with the comments raised by me (editor) and reviewer 1. A major revision and detailed point by point response is required prior to publication. Also respond to the editor's comments raised during revision 1.

Authors should give proper explanation of results, or why they found inconsistent patterns in the data. Results and Discussion section should be thoroughly improved. Present your key findings in the conclusion section. Change the description of treatments as well-watered’, ‘moderate’ and ‘severe’ drought. Title should also be improved. Replace number (10) with text (ten). Authors should also carefully improve the language of the draft, to omit the minor grammatical and typo mistakes.

Reviewers' comments:

Reviewer's Responses to Questions

**Comments to the Author**

1. If the authors have adequately addressed your comments raised in a previous round of review and you feel that this manuscript is now acceptable for publication, you may indicate that here to bypass the “Comments to the Author” section, enter your conflict of interest statement in the “Confidential to Editor” section, and submit your "Accept" recommendation.

Reviewer #1: (No Response)

Reviewer #2: All comments have been addressed

2. Is the manuscript technically sound, and do the data support the conclusions?

Reviewer #1: No

Reviewer #2: Yes

3. Has the statistical analysis been performed appropriately and rigorously? 

Reviewer #1: No

Reviewer #2: Yes

4. Have the authors made all data underlying the findings in their manuscript fully available?

Reviewer #1: Yes

Reviewer #2: Yes

5. Is the manuscript presented in an intelligible fashion and written in standard English?

Reviewer #1: No

Reviewer #2: Yes

6. Review Comments to the Author

Reviewer #1: General

I think that perhaps my comments were not clear on the first revision. The structure of the results/discussion needs to be significantly improved as it does not attempt to explain your results in depth or detail. The first paragraph of each section in the results/discussion is generic and not related to explaining/interpreting your results. Therefore, this information belongs in the introduction where it is required to justify your approach. So, for each section of the discussion, re-locate the first paragraph to the introduction. But please ensure that the introduction is not a ‘cut and paste’ disjointed section as a result, it needs to flow and be logical. The remaining paragraphs in each section of the results/discussion have largely not changed since my first review. For the majority, it seems pointless to rank species within treatment --- these paragraphs are largely unreferenced and the reader is confused by you continually ranking species. Save the ranking for your strongest indicator of tolerance which is the change in each response variable from well-watered to severe drought conditions. That should be the focus and first paragraph of each section. Indeed, there really only needs to be one decent paragraph in each section. Species showing the most dramatic change from well-watered to droughted conditions are clearly the most sensitive, therefore this is the only ranking you require. Do this for each variable, then for the combined variable at the end. Comment then on the relative importance of each variable in section 4.6 --- for instance, does the ranking produced by the membership function relate well to any individual measure, and why? I hope this is clear. I have provided detailed comments for section 4.2 below to help, but please apply this logic throughout your paper. With regard to other general comments, it is very, very important that in a prominent position early in your results and discussion that you comment on the short length of exposure to drought AND (as you state in the first sentence in Section 3.5) that the sampling was done only 10 days after transplant. It is highly likely these two factors affected the outcome of your experiment and people will be highly critical of the short duration of exposure to experimental conditions unless you address it early. For example, sampling only 10 days after planting, it is highly likely that roots did not develop and therefore manipulation of soil moisture is largely irrelevant. The results you observed may simply be transplant shock with such a short timeframe. Comment must be made on these issues if people are to have confidence in the results.

Section 4.1

There is inconsistency between how soil water content is discussed --- e.g. in the methods, it is referred to as relative water content (RWC), then as soil moisture content in the subsection heading for Section 4.1, then as soil water content in the first sentence of that section. Please be consistent with terminology throughout the paper.

Section 4.2

Paragraph 1: is not very good because it does not talk about your data/your study first, which is what the results/discussion is for --- the focus needs to be on your data, using the literature to support. The background about why measuring membrane permeability is useful for quantifying drought tolerance is all very useful information, but it belongs in the introduction --- not the results/discussion. My suggestion is that, given the introduction is so short, and that the results/discussion is so long, these ‘general information’ paragraphs in each section of the results/discussion should be moved to the introduction of the paper. Still on the example of section 4.2, your first paragraph would be the one currently under Figure 1 (the current paragraph 2).

Paragraph 2: In the second paragraph, we are told that MP values under well-watered conditions reflect normal/healthy status. Why then do you rank species by well-watered MP in the next sentence? This is not logical to me --- if you have a healthy, well-watered treatment group, then the MP value is “ideal” --- therefore the only relevant thing to talk about here is how MP changes when you impose drought stress (i.e., the last paragraph in this section). For instance, are well-watered D.barbatus plants less drought tolerant because they have very high MP under well-watered conditions? Not if you consider a well-watered D.barbatus plant to be healthy…What I am saying here is your focus should be on looking at the change within species, from well-watered to droughted conditions to define drought tolerance. There should be only one species ranking based on MP and I would do this by starting with the least tolerant (the species which died), then rank the remainder based on the magnitude of the change in MP between well-watered and droughted. The remaining discussion on who changed from moderate to severe is not required.

Paragraph 3: Similarly, you focus the third paragraph on C. bipinnata --- you suggest that the high MP for well-watered C. bipinnata means that it is not tolerant to adversity. Firstly, “adversity” is not specific, you conducted a drought experiment, so be specific and refer to drought --- but secondly, your opening sentence in paragraph two (which is unreferenced, by the way…and needs to be!) states clearly that MP values under well-watered conditions reflect “healthy”…therefore, you cannot say that the high MP value for C. bipinnata implies a lack of tolerance because this value is taken from the well-watered treatment. On the other hand, the fact that this species died in response to both moderate and severe drought is good evidence of a higher sensitivity. You refer to the study by Wu et al (2018) who found the same as you ---- I thought you said in the introduction that the drought tolerance of these species has not been quantified before, therefore that statement is also incorrect and needs to be changed. BUT, more to the point, WHAT did Wu et al (2018) say about this species, did they come up with an explanation for high MP? And more to the point, what is your explanation for it? For each result, I am looking for an explanation --- e.g. the last sentence in the last paragraph of section 4.2 --- here, you make a good link between your two sensitive species and findings from the literature with regard to root growth --- but your treatment was very short, therefore it likely did not cause poor root development, but instead may indicate these plants have inherently superficial/poor root systems….this is good discussion and this is what I need to see for each result you present --- not only who was bigger/smaller etc., but why you think that might be the case. Every main result needs to be explained.

Paragraph 4: Contains no references and is largely unnecessary --- ranking species here is pointless.

Paragraph 5: Again, the ranking within treatment is largely unnecessary, the CHANGE from well-watered to droughted is relevant. The reference is generic about MP and doesn’t explain your results.

Paragraph 6: The reference is again generic and the first two sentences should have been used earlier in the discussion. This paragraph should be the main focus of section 4.2 and requires expansion to discuss not only what happened (species ranking), but further, more in-depth assessment of why. I know that your motivation for this paper is to rank species by drought tolerance, but we need a good explanation of the observed trends in the data, using the literature, before we can utilise/have confidence in that ranking. Considering this, have you thought about making section 4.6 the first section in your results/discussion? This is the ultimate ranking of species drought tolerance, so why not just make a ranking here, then you can explore the detail of the constituent components of the contributors to the integrated variable, each of which is ‘less interesting’ than the overall aim of your paper. This is using the logic of the newspaper article, where the most important information comes first, and the least important/interesting comes last.

7. PLOS authors have the option to publish the peer review history of their article (what does this mean?). If published, this will include your full peer review and any attached files.

Reviewer #1: No

Reviewer #2: No

---

## [Author Response · Author response to Decision Letter 2]

19 May 2020

Response to Reviewers

 We authors appreciate Reviewer #1 and editor for your detailed guide to further revise the manuscript. We have renewed the content following your suggestions.

Summary of Reviewer #1’s viewpoints:

 Reviewer #1 took the section 4.2 as an example and showed us how to organize the article structure. 

 According to the suggestions, we made the following major changes in Results and Discussion: 

1. Move the first paragraphs in "Result and Discussion" to the “Introduction" section. 

2. For each parameter, statistically significant changes were described instead of detailed species data lists.

3. We focued on the important phenomenon and their scientific explanation. 

4. We summarised the major finding points in the last paragraph for MP, SOD and POD. 

In addition, we corrected the spelling mistakes , grammatical and other errors.

---

## [Decision Letter · Decision Letter 3]

2 Jun 2020

Drought Resistance of 10 Ground Cover Seedling Species during Roof Greening

PONE-D-19-20236R3

Dear Dr. Liu,

We’re pleased to inform you that your manuscript has been judged scientifically suitable for publication and will be formally accepted for publication once it meets all outstanding technical requirements.

Kind regards,

Saddam Hussain

Academic Editor

PLOS ONE

Reviewers' comments:

Reviewer's Responses to Questions

**Comments to the Author**

1. If the authors have adequately addressed your comments raised in a previous round of review and you feel that this manuscript is now acceptable for publication, you may indicate that here to bypass the “Comments to the Author” section, enter your conflict of interest statement in the “Confidential to Editor” section, and submit your "Accept" recommendation.

Reviewer #1: All comments have been addressed

2. Is the manuscript technically sound, and do the data support the conclusions?

Reviewer #1: Yes

3. Has the statistical analysis been performed appropriately and rigorously? 

Reviewer #1: Yes

4. Have the authors made all data underlying the findings in their manuscript fully available?

Reviewer #1: Yes

5. Is the manuscript presented in an intelligible fashion and written in standard English?

Reviewer #1: Yes

6. Review Comments to the Author

Reviewer #1: Dear Authors,

Thanks for your efforts in addressing my comments. I am happy with the revisions made now, the paper is much improved.

7. PLOS authors have the option to publish the peer review history of their article (what does this mean?). If published, this will include your full peer review and any attached files.

Reviewer #1: No

---

## [Editor Report · Acceptance letter]

10 Jun 2020

PONE-D-19-20236R3 

Drought Resistance of 10 Ground Cover Seedling Species during Roof Greening 

Dear Dr. Liu:

I'm pleased to inform you that your manuscript has been deemed suitable for publication in PLOS ONE. Congratulations! Your manuscript is now with our production department. 

Kind regards, 

on behalf of

Dr. Saddam Hussain 

Academic Editor

PLOS ONE